# Self-Assembled Peptide Habitats to Model Tumor Metastasis

**DOI:** 10.3390/gels8060332

**Published:** 2022-05-25

**Authors:** Noora Al Balushi, Mitchell Boyd-Moss, Rasika M. Samarasinghe, Aaqil Rifai, Stephanie J. Franks, Kate Firipis, Benjamin M. Long, Ian A. Darby, David R. Nisbet, Dodie Pouniotis, Richard J. Williams

**Affiliations:** 1School of Health and Biomedical Sciences, RMIT University, Melbourne, VIC 3000, Australia; noora.albalushi83@gmail.com (N.A.B.); iandarby1@gmail.com (I.A.D.); 2School of Engineering, RMIT University, Melbourne, VIC 3000, Australia; mboydmoss@gmail.com (M.B.-M.); aaqil.rifai@deakin.edu.au (A.R.); kate.firipis@gmail.com (K.F.); 3School of Medicine, Deakin University, Waurn Ponds, VIC 3216, Australia; r.samarasinghe@deakin.edu.au; 4BioFab3D, Aikenhead Centre for Medical Discovery, St. Vincent’s Hospital, Fitzroy, Melbourne, VIC 3065, Australia; 5Institute for Mental and Physical Health and Clinical Translation, Deakin University, Waurn Ponds, VIC 3216, Australia; 6Laboratory of Advanced Biomaterials, Research School of Chemistry and the John Curtin School of Medical Research, Australian National University, Canberra, ACT 2601, Australia; stephanie.franks@anu.edu.au (S.J.F.); david.nisbet@unimelb.edu.au (D.R.N.); 7Future Regions Research Centre, Federation University Australia, Mount Helen, VIC 3353, Australia; bm.long@federation.edu.au; 8The Graeme Clark Institute, The University of Melbourne, Melbourne, VIC 3010, Australia; 9Department of Biomedical Engineering, Faculty of Engineering and Information Technology, The University of Melbourne, Melbourne, VIC 3053, Australia; 10Melbourne Medical School, Faculty of Medicine, Dentistry and Health Science, The University of Melbourne, Melbourne, VIC 3010, Australia

**Keywords:** self-assembly, peptide, hydrogel, functionalisation, cancer, matrix

## Abstract

Metastatic tumours are complex ecosystems; a community of multiple cell types, including cancerous cells, fibroblasts, and immune cells that exist within a supportive and specific microenvironment. The interplay of these cells, together with tissue specific chemical, structural and temporal signals within a three-dimensional (3D) habitat, direct tumour cell behavior, a subtlety that can be easily lost in 2D tissue culture. Here, we investigate a significantly improved tool, consisting of a novel matrix of functionally programmed peptide sequences, self-assembled into a scaffold to enable the growth and the migration of multicellular lung tumour spheroids, as proof-of-concept. This 3D functional model aims to mimic the biological, chemical, and contextual cues of an in vivo tumor more closely than a typically used, unstructured hydrogel, allowing spatial and temporal activity modelling. This approach shows promise as a cancer model, enhancing current understandings of how tumours progress and spread over time within their microenvironment.

## 1. Introduction

The local environment of a tumour is a complex interplay between cancerous, precancerous and stromal cells within a specific extracellular matrix (ECM). Interactions between the ECM and these cells form an intricate, dynamic microenvironment, also known as the tumour microenvironment (TME), which has a significant influence on tumour growth and progression [1]. Identifying the importance of the TME and the components present in these ecosystems has recently become an important field in cancer research, as it has been shown that a single alteration to any one of these components can transform the tumour and the microenvironment in which it resides [2]. Understanding this system is vital for insights to tumorigenesis, progression, and metastasis, motivating the use of bench-top culture models in an effort to replicate the surrounding in vivo microenvironment of a tumor. However, there is still a requirement to fabricate a more comprehensive biomimetic 3D model. The current gold standard for in vitro cancer studies is limited by growing cancer cells on 2D cell culture surfaces—such as tissue culture substrates or glass coverslips. These conventional 2D cell culture models are simple, cheap, and accessible; they have significantly enhanced our understanding of cancer biology. However, 2D models fail to present architectural physical and chemical signals found in the tumour niche, and as such, the role of the in vivo TME is dismissed. Specifically, 2D cell culture surfaces lack the direct, multidimensional cell-cell and cell–matrix interactions that govern major cellular processes, which can have a significant impact on developing effective cancer therapies [3,4].

The natural TME is a dynamic, complex system, consisting of the ECM, cancer cells, blood vessels, and a host of cell types from the stroma—such as adipocytes, fibroblasts, immune and endothelial cells—along with an abundance of soluble factors such as cytokines and growth factors [5]. An important parameter for understanding cancer progression is the phenotypic changes in the cells of the stroma, resulting in extensive ECM remodeling that cannot be reproduced in 2D monocultures [6]. Recently, the formation of tumour spheroids has emerged as a popular technique to model aspects of the 3D cancer in vitro [7]. They are similar to in vivo solid tumours: Both display an oxygen gradient and nutrient distribution; localized cell-cell communication; and distinct peripheral proliferation and centralized necrotic zones (Figure 1). Recently, research has focused on the development of highly-efficient, cancer spheroid formulation for drug screening applications [8]. However, from a bioengineering perspective, the major limitation of spheroid models is a lack of relevant and receptive external ECM support, with spheroids typically studied within an attachment-free microenvironment that is inherently deficient of relevant mechanical and biochemical characteristics [9].

One of the main requirements for modelling in vivo tumour systems in vitro is the existence of a suitable biomimetic scaffold to dynamically support cell-cell and cell-matrix interactions [10]. The ECM of the TME is composed of a complex network of biopolymer fibres, consisting of glycoproteins, collagens, glycosaminoglycans, proteoglycans and proteins, that communicate with, support and hydrate the cells [11]. This raises the potential to replicate the complexity of the in vivo TME through the incorporation of ECM components; however, they need to be synthetically reproduced. Hydrogel scaffolds are excellent candidates for this application, as they consist of an interpenetrating mesh of highly hydrated fibres. Recently, hydrogels have been used to bioprint advanced tumour models—allowing for the precise placement of living cells, functional biomaterials, and programmable drug-release capsules [12,13].

A diverse range of natural and synthetic biomimetic scaffolds, typically in the form of hydrogels, has attracted attention as candidate in vitro 3D tumour models [14]. For example, both patient-derived tumour organoids [15,16] and tumour spheroids [17] have been embedded within an easily handled tumour-derived hydrogel, commercially available as Matrigel, or decellurlarised tissue [18]. The major drawbacks, however, are methodological complexities and the lack of reproducibility [19]; spheroids embedded within animal-derived Matrigel were limited due to problems with immunogenicity, handling, useability and batch-to-batch variation arising during manufacturing [20]. Recently, Baker et al. have investigated the use of a new oxime-crosslinked hyaluronan (HA) hydrogel for breast-cancer modelling, finding improved cell response to the Rac inhibitor (EHT-1864) and the PI3K inhibitor (AZD6482) when cultured in HA-oxime versus Matrigel [21]. A simple approach to 3D cancer cell culture is embedding cancer cells in suspension within plant-derived hydrogels [22]. While these hydrogels allow for the simple encapsulation and recovery of cells, they are limited in terms of sophisticated TME mimicry; they offer poor control over the morphology of structures formed during gelation and a lack of functionality (without chemical modification) and, as such, they inherently lack cell-adhesion sites [23]. There is clearly a need for synthetic hydrogels that are chemically well-defined, and easily programmed with ECM elements.

Recently, significant focus has centred on simple peptides that can self-assemble to form sophisticated scaffolds with advanced biological functions [24]. These self-assembling peptides (SAPs) have emerged as promising biomaterials for application in cancer therapy [25]. The self-organising nature of these materials provides strategic advantage, allowing for triggered assembly under specific, targeted conditions. Consequently, SAPs have been used as novel inhibitors of cancer development through alkaline phosphatase (ALP)-mediated self-assembly [26,27]. However, SAP networks have also seen tremendous use as ECM mimics in regenerative medicine, owing to their ability to form highly biomimetic structures replete with biologically relevant motifs [28]. Thus, recent research has highlighted the potential of these materials for use as artificial TMEs, allowing for the generation of advanced tumour models, and subsequently enabling superior analysis of prostate tumour progression in vitro [29].

A promising class of SAP hydrogels is Fluorenylmethyloxycarbonyl-SAPs (Fmoc-SAPs). Fmoc-SAPs are SAP sequences capped with an Fmoc aromatic group at the N-terminal. This Fmoc group drives the self-assembly process through π-π stacking between aromatic groups, followed by formation of nanofibres through non-covalent interactions. These assembly processes result in nanofibres with epitope bioactivity decorating their external surface [30,31]. The decorated nanofibres intertwine longitudinally to form an entangled matrix hydrogel with similar mechanical properties, physical properties, and epitope presentation to the in vivo ECM (Figure 2) [30]. Fmoc-SAPs are advantageous because they are easily synthesised, biocompatible, and have the capacity to be functionalised with specific bioactive peptide sequences at high density on the surface of the nanofibres [32]. Utilising these materials, we present an enhanced in vitro 3D lung tumour model, which we demonstrate to have significant relevance to the in vivo TME. Multicellular spheroids of murine Lewis lung cancer cells (LLC) and murine fibroblast cells (NOR-10, derived from skeletal muscle) were encapsulated within self-assembled pro-adhesive peptide sequences from ECM components. Peptide sequences include isoleucine–lysine-valine–alanine–valine (IKVAV) derived from laminin, a component found in ECM basement membrane and arginine–lycine–aspartate (RGD) derived from fibronectins, components of connective tissue [33]. These peptide sequences have been used widely to engineer scaffolds for 3D cell culture systems, and we have previously shown that IKVAV-peptide sequences induce cellular adhesion and tubule formation [34] and RGD-peptide sequences enhance interaction between integrins on the cells and biomaterial [35,36]. In order to compare our nanostructured hydrogel to a control with broadly similar bulk properties but lacking the programmed chemical richness of our peptide system, we compared it to a commonly used agarose-based hydrogel.

The microstructures of the systems were analysed via TEM and Cryo-SEM. Small angle X-ray scattering (SAXS) was used to determine the relative size of the mesh presented by the scaffold features. In order to probe the efficacy of the 3D model, we compared outcomes from cells grown within one of three microenvironments, 2D (a monolayer cells grown on flat tissue culture plates), 2.5D (spheroid on the bottom of the well plate, surrounded by media) and 3D (spheroid encapsulated within hydrogel) (Figure 1D). Cells in the various systems were monitored for metabolic activity, adhesion, cytoskeleton organisation and fibroblast activation. Our data indicated that cultured lung cancer spheroids within an Fmoc-SAP hydrogel (3D) presents a significant improvement on 2D and 2.5D models in the terms of: (1) significantly enhanced overall metabolic activity of co-cultures; (2) promoted cell invasiveness as demonstrated by reduced vinculin levels; (3) increased migratory phenotype of cells as demonstrated by actin cytoskeleton reorganisation; and (4) acquired a mesenchymal phenotype (increased alpha smooth muscle actin (α-SMA) expression). We suggest that this model provides a highly promising platform to further investigate cancer biology, cellular processes and the in vitro response to anti-cancer therapeutics.

## 2. Material and Methods

Fmoc-FRGDF and Fmoc-DIKVAV were purchased from PepMic (Suzhou, China) with purity >95% and desalted. LLCs were maintained in Dulbecco’s Modified Eagle’s Medium (DMEM) (Life Technologies, Melbourne, Australia) supplemented with L-Glutamine, D-Glucose (1 g/L), 10% (*v*/*v*) fetal bovine serum (FBS), Sodium Pyruvate (110 mg/mL), and 1% (*v*/*v*) PenStrep (termed as complete 10 media). NOR-10 fibroblasts were maintained in Dulbecco’s Modified Eagle’s Medium (DMEM) supplemented with L-Glutamine, D-Glucose (1 g/L), 20% (*v*/*v*) FBS, Sodium Pyruvate (110 mg/mL), and 1% (*v*/*v*) PenStrep (termed as complete 20 media).

### 2.1. Co-Assembled Fmoc-FRGDF/Fmoc-DIKVAV Hydrogel Preparation

A co-assembled Fmoc-SAP hydrogel was prepared by mixing equal mass of Fmoc-FRGDF and Fmoc-DIKVAV peptides to a total mass of 10 mg as described previously [37]; however, once a pH of 7.4 was achieved, complete medium was added into the hydrogel solution to make the gel up to 1 mL instead of PBS solution. The SAP hydrogel was then exposed to UV light in the tissue culture hood for a period of no less than 2 h for sterilisation purposes before cell seeding. In rheological analysis, 1xDPBS was added in place of media. In electron microscopy sections, deionised H_2_O (dH_2_O) (MilliQ, Merck Millipore, Bayswater, Australia) was substituted in place of media.

### 2.2. Agarose Preparation

A 2% low melting agarose (#200-0030, Progen Industries Limited, Toowong, Australia) solution was made in dH_2_O and melted by heating in the microwave with continuous rapid stirring until the solution came to a boil and then autoclaved. Sterile 20% agarose was allowed to cool to 37 °C before mixing with an equal volume of sterile warm complete DMEM (with 10% or 20% FBS) to form a 1% agarose-medium mixture. The mixture was allowed to set at room temperature before spheroid encapsulation.

### 2.3. Cryo-Scanning Electron Microscopy (CryoSEM)

FEI Quanta 200 environmental scanning electron microscope in CryoSEM mode at an operation voltage of 15 kV was used to image samples. Hydrogel samples were prepared as described with dH_2_O in place of media and were placed into a CryoSEM sample holder before being plunged into a liquid nitrogen (LN_2_) slosh to snap freeze samples and avoid ice crystal formation. The sample was then placed into the sample preparation chamber (−180 °C, high-vacuum) and cracked open using a blade. The cracked sample was then sublimated in the microscope using a heater at −90 °C for 4 min. The sample was then gold sputter coated in the preparation chamber to avoid sample charging. Finally, samples were transferred back to the microscope chamber and positioned under the detector for imaging.

### 2.4. Small-Angle X-ray Scattering (SAXS)

SAXS was performed at the Australian Synchrotron (Melbourne, Australia) using the SAXS/WAXS beamline. Measurements were acquired at a calibrated camera length of 967.667 mm with X-ray energy of 12 KeV (1.03320 Å); allowing for the scattering vector (q) to be measured across the range of 0.018 to 0.92 Å^−1^. The diffraction pattern was recorded on a Pilatus 1M detector (169 mm × 179 mm, effective pixel size (172 μm × 172 μm) and processed using the Australian Synchrotron ScatterBrain Software (Melbourne, Australia). Hydrogels were prepared as detailed above and loaded into 1.5 mm glass capillaries with a wall thickness of 0.1 mm (Hilgenberg, GmbH, Malsfeld, Germany). PBS backgrounds were collected before samples were loaded. Each sample (and background) was subjected to ten, 1 s^−1^ exposures at multiple points along the capillary to minimise sample burning. Repeat measurements were summed using Scatterbrain and q calibrated using an AgBeh sample, and intensity was normalised and set on an absolute scale using water and air shots. Due to poor scattering, backgrounds were scaled by 0.9 before subtraction from the sample scattering data. For co-assembled Fmoc-SAP hydrogel fibril radius calculations, data were subject to indirect Fourier-transform (IFT) analysis and P(r) inversion using SASView (SASView, Australia) to calculate the average diameter of the fibrils in the sample. For agarose samples, data were fit to shape independent models using SASView. Specifically, A single power law model and mesh size was determined by fitting with a two-power law model at low-mid q range (0.018–0.2 Å^−1^). All acquired scatter curves, and subsequent fits were further graphed using GraphPad Prism 7 (GraphPad Prism Software Inc., San Diego, CA, USA) for ease of graphical visualisation.

### 2.5. Cell Culture: LLC and NOR10

Murine Lewis Lung Carcinoma (LLC) cell line was kindly provided from Peter MacCallum Cancer Centre, Melbourne, Australia, and the murine skeletal muscle fibroblast (NOR-10) cell line was purchased from ATCC, USA. The LLC cell line was maintained in complete 10 media. NOR-10 murine fibroblasts were maintained in complete 20 media. Both cell lines were incubated at 37 °C and 5% CO_2_. After reaching 80–90% confluence, LLC cells were detached using 0.05% Trypsin/EDTA and NOR-10 using 0.25% Trypsin/EDTA.

### 2.6. Culture Conditions in 2D, 2.5D and 3D Spheroids

LLC and NOR-10 were maintained as monolayer cultures in the media described above. To generate spheroids, LLC, NOR-10 and co-culture spheroids (LLC and NOR-10 at 1:1 ratio) were seeded at 10 × 10^3^ cells/well using ultra-low attachment 96-well round-bottomed plates (Corning #7007). Over 24 h of incubation, dense and circumscribed multicellular spheroids were observed under a bright field inverted microscope. These spheroids were termed 2.5D. The 3D spheroids were formed by encapsulating the formed 2.5D spheroids with co-assembled Fmoc-SAP hydrogel. Spheroid encapsulation was considered complete when the hydrogel was observed surrounding the spheroid. For cultures in 2D, cells were cultured in common tissue culture 96-well flat-bottomed plates (Corning #3595) using the same culture media and at the same cell concentration. Only for flow cytometry, 3D spheroids were harvested from the hydrogels by gently pipetting via sterile P200 pipette tip and were used for flow cytometry as described in the Materials and methods sections.

### 2.7. Metabolic Activity

The metabolic activity of LLC, NOR-10 and co-culture 3D spheroids in response to encapsulation within Fmoc-SAP hydrogel was determined with CellTiter 96^®^ AQueous One Solution Cell Proliferation Assay (Promega, Madison, WI, USA) and compared to 2.5D spheroids, according to manufacturer’s instructions. Briefly, after 3D and 2.5D spheroid formation and incubation for a pre-determined time, MTS solution was added directly to each well followed by incubation at 37 °C for 4 h. Then, 100 μL of supernatant was transferred to a clean 96-well flat bottom plate and cells metabolic activity was determined by measuring the absorbance at 490 nm using a plate reader (Clariostar plate reader, BMG LABTECH, Mornington, Australia).

### 2.8. Flow Cytometry

After culturing LLC, NOR-10 and co-culture 3D, 2.5 spheroids and 2D cultures for 72 h, vinculin and α-SMA expression levels were determined by flow cytometry. Harvested 3D, 2.5D spheroids and cells in 2D cultures were trypsinised with 0.25% trypsin-EDTA. Spheroids were also mechanically dissociated by repeated pipetting. Then, single dissociated cells were fixed with 2.5% paraformaldehyde for 20 min, permeabilised with 0.5% saponin in PBS, and blocked with 0.5% BSA in PBS. Immunostaining was performed by incubating the cells with either mouse monoclonal anti-vinculin primary antibody (1:100 dilution, ab18058, Abcam) followed by Alexa Fluor 594-conjugated secondary antibody (1:200 dilution, A11005, Goat Anti-Mouse IgG (H+L); Life Technologies Pty Ltd., Welshpool, Australia); or mouse monoclonal anti-α-SMA primary antibody (1:100 dilution, ab7817, Abcam) followed by Alexa Fluor 488-conjugated secondary antibody (1:200 dilution, A11059, Rabbit Anti-Mouse IgG (H+L); Invitrogen, Carlsbad, CA, USA). After incubation, cells were washed with PBS and resuspended in 500 μL FACS (Fluorescence-activated cell sorting) Fix buffer ready for flow cytometry. Negative controls (incubation in secondary antibody only) were prepared for each experimental condition. Flow cytometry analysis was performed on a FACS Canto™ II Flow Cytometer (BD Biosciences, San Jose, CA, USA). Data were analysed using the Flowing software 2 (v2.5.1, University of Turku, Turku, Finland). Each experiment was performed at least 3 times in duplicate.

### 2.9. Cell Migration Tracking

Prior to co-culture and spheroid formation, cells were fluorescently labelled according to manufacturer’s instructions (LLC cells (CellTracker™ Red CMTPX Dye, C34552, Invitrogen); NOR-10 cells (MitoTracker Green FM, M7514, Invitrogen)). Briefly, cell concentrations were adjusted to 1 × 10^6^ cells/mL in PBS, with CMPTX red and MitoTracker Green added at a final working concentration of 5 μM and 10 μM, respectively. Cells were incubated for 30 min at 37 °C whilst protected from light, followed by fluorescence quenching with two volumes of complete media and washed three times. Fluorescently labelled cells were co-cultured into spheroids and encapsulated within either co-assembled Fmoc-SAP hydrogel or 1% agarose gel. Cell migratory behaviour was tracked by confocal microscopy after 72 h of incubation.

### 2.10. Immunofluorescence Staining for Vinculin and F-Actin

F-actin and vinculin co-localisation was assessed in 2D, 2.5D and 3D co-cultures. After 72 h of incubation, cells were fixed in 4% paraformaldehyde and then washed twice in 0.05% Tween-20 in PBS wash buffer, before permeabilisation with 0.5% Triton X-100 solution in PBS blocked with 1% BSA in PBS. Then, cells were stained with mouse monoclonal anti-vinculin primary antibody (1:100 dilution, ab18058, Abcam) at 4 °C overnight. Following washing with wash buffer, spheroids were incubated with Alexa Fluor 488-conjugated secondary antibody (1:200 dilution, A11059, Rabbit Anti-Mouse IgG (H+L); Invitrogen, Carlsbad, CA, USA) and tetramethylrhodamine isothiocyanate (TRITC)-conjugated phalloidin (1:200, R415; Life Technologies Pty Ltd., Welshpool, Australia). Finally, cell nuclei were counterstained with DAPI (R37606, Life Technologies, and Grand Island, NY, USA) for 5 min and washed with wash buffer. Fluorescence images of the cells were captured with a laser scanning confocal microscope (LSCM) (objective 40×, excitation wavelengths: 488, 405 and 561 nm; A1R+ confocal microscope system; Nikon, Tokyo, Japan). Prior to imaging, spheroids were immersed in PBS to avoid drying out.

### 2.11. Statistical Analysis

All statistical analysis was conducted using GraphPad Prism 7.0 (Prism v7.0, GraphPad Software, San Diego, CA, USA). Data is presented as mean ± standard error of the mean (SEM). Two groups were compared using student’s two-tailed unpaired *t*-test and multiple groups were compared using Two-Way Analysis of Variance (ANOVA) with post-hoc analysis using Tukey’s multiple comparison tests. *p*-value < 0.05 was considered statistically significant. All experiments were performed at least 3 times in duplicate. SAXS data were presented as calculated value ± uncertainty as determined with SASview (SASView, Sydney, Australia).

## 3. Results and Discussion

### 3.1. Comparison of Hydrogel Networks

Initially, we compared the topography of the nanofibrous peptide-rich Fmoc-SAP network to an agarose control. Agarose was selected as a control due to its similar macroscale properties with a comparatively inert nature (Section 2.4). A combination of negative stain Transmission Electron Microscopy (TEM), Cryogenic-Scanning Electron Microscopy (CryoSEM) and Small Angle X-ray Scattering (SAXS) techniques was employed to ensure validity of the agarose control when used in conjunction with the Fmoc-SAP network (Figure 2). These techniques allowed us to analyse the nano and microstructures of the systems.

As expected, the microscopy showed that the Fmoc-SAP produced a nanofibrous morphology that underpinned a microscale porous scaffold as previously reported (Figure 2Ai,iii). Fmoc-FRGDF/Fmoc-DIKVAV hydrogels formed a dense network of fibrous structures that ranged in size, with the larger fibers likely a result of fiber bundling, in accordance with previous findings [34]. The cylindrical nature of the fibres was confirmed by SAXS power law analysis (transition from q^−4^ at mid q-range trending to q^−1^ at low q-range) and the average fibre radius (7.3 nm) was determined by Indirect Fourier Transform (IFT) of the SAXS data [38] By microscopy, the agarose control presented a similar microstructure to the Fmoc-SAP hydrogel; however, it showed no such nanofibrillar organisation and thus was not suited to the SAXS IFT analysis (Figure 2Aii,iv). However a power law analysis of the scattering at low q revealed a branched polymer network type conformation (Power = 2.50, Appendix A) [39]. Both hydrogels demonstrated a highly dense network of high porosity, but in the agarose system, the network appears more chaotic with thicker bundles and large clump-like intersections evident. However, analysis of mesh size by SAXS power law crossover revealed a more porous nanoscale network in the Fmoc-SAP gel (10.2 nm vs. 19.4 nm for the agarose sample, Appendix A) [40]. Both structures have similar macroscopic properties and microscale structure but are underpinned with a different nanoscale and molecular structure, which makes agarose an ideal control for the 3D culture.

### 3.2. Evaluating Metabolic Activity of Lung Cancer and Stromal Cells Spheroids

The initial aim from a biological standpoint was then to compare the effect of the functionalised Fmoc-SAP hydrogel on metabolic activity of tumour cells and stromal cells. Metabolic activity provides a general screen of cell health; hence, for this purpose, murine LLC and NOR-10 mono-spheroids and co-culture spheroids (1:1 ratio) were encapsulated within Fmoc-FRGDF/Fmoc-DIKVAV. The spheroids cultivated within the Fmoc-SAP hydrogel were termed 3D spheroids and their metabolic activity was compared to a control group of non-encapsulated spheroids (termed 2.5D spheroids) using an MTS assay over a time period of 72 h. Spheroids that were 3D and 2.5D were either cultured with LLC, NOR-10 or co-cultured with both LLC and NOR-10. At 18 to 24 h post-culture, we noted that the metabolic activity of LLC, NOR-10 and co-culture 3D spheroids was significantly enhanced compared to their 2.5D counterparts (LLC *p* < 0.0001, NOR-10 *p* = 0.0125, co-culture *p* = 0.0004, Figure 2C). At 72 h post-culture, apart from LLC 3D spheroids, a similar trend was observed with NOR-10 and co-cultured 3D spheroids compared to 2.5D spheroids (NOR-10 *p* < 0.0001, co-culture *p* < 0.0001). Our results highlight that growing lung cancer spheroids in a 3D environment in vitro in the presence of two adhesion peptide sequences is effective, as it significantly enhanced the overall metabolic activity of 3D co-culture spheroids compared to 2.5D cultures over 72 h. This observation proves that Fmoc-FRGDF and Fmoc-DIKVAV peptide sequences provide spheroids with a more beneficial microenvironment that is not only biocompatible but also promotes cellular growth when compared to spheroid culture (either of individual or co-cultured cells). We have previously shown that our tailored Fmoc-SAP hydrogel is biocompatible and supported the viability of human mammary fibroblast cells (hMFCs) after seeding in Fmoc-FRGDF hydrogel over 5 days [36]. Similarly, Rui et al. showed that Fmoc-FRGDF supported the viability of oral tongue squamous carcinoma cells (SCC25) for 3 days by MTS assay [41]. Our results are also consistent with a previous study that observed a significantly enhanced proliferation rate of epithelial ovarian cancer cell (OV-MZ-6) spheroids embedded within an RGD-functionalised polyethylene glycol (PEG)-based hydrogel, compared to non-RGD-functionalised hydrogel and conventional 2D cultures [42]. Collectively, these results emphasize the inherent biocompatibility of Fmoc-SAP hydrogels for lung cancer spheroid growth.

Conversely, other cell lines, such as human submandibular salivary gland (HSG) cells on Matrigel and colorectal cancer (CRC) cell lines on laminin-rich extracellular matrix (lrECM), have shown a reduction in the overall metabolic activity of cells cultured in 3D as opposed to 2D [43,44]. It has been speculated that the difference in proliferation rate of cell lines between 3D and 2D cell cultures might be due to altered gene expression [44]. However, these examples utilised matrices that were very different to our functionalised hydrogel, indicating that the reduction in proliferation rate in 3D may also depend on the suitability of the chosen culture matrix to a 3D environment, and the cells’ affinity for the matrix in 3D. Overall, the metabolic activity rate of cells grown in 3D cell culture more closely represents the growth of cells in vivo than those cultured in an artificial 2D cell culture [45]. Our results underline the significance of the cell-ECM interaction on cellular behaviour and responses to the TME.

### 3.3. Enhanced Spheroid Adhesion within Functionalised Fmoc-SAP Hydrogel

Vinculin is a cytoskeleton protein contributing to cell-cell adhesion through cadherin and cell-ECM connection through integrin receptors [46,47]. It is closely associated with a role in governing cell-matrix adhesion. It plays a critical role in regulating integrin clustering, and generates the forces required to attach the cells to the ECM. A significant reduction in vinculin levels eventually results in decreased cell adhesion and induction of motility, and it can be considered a mechanism for monitoring cancer cell invasion and metastasis [46,47]. The down-regulation of vinculin was reported in many invasive cancers including colorectal cancer, melanoma, vaginal and cervical carcinoma [47,48,49]. Therefore, to explore the functional association between cancer cell adhesion and ECM, vinculin expression in 3D mono- and co-culture spheroids was compared to 2.5D and 2D cell culture by flow cytometry (Figure 3A and Appendix A). Our results showed that at 72 h post culture, vinculin was significantly downregulated in 3D NOR-10 (0%) and co-culture (0%) spheroids compared to 2D cell cultures (26.3% and 24.1%, respectively) (NOR-10 *p* = 0.0004, co-culture *p* = 0.0019). In 2.5D spheroids, there was no significant difference in the percentage of cells expressing vinculin (10.3% LLC, 9.8% NOR-10 and 11.6% co-cultured spheroids) compared to 3D and 2D spheroids (ns = not significant, *p* ≥ 0.05).

In line with our results, several studies have reported low levels of vinculin in metastatic colorectal cancer tissue samples and metastatic squamous cell carcinomas [47,50]. Li et al. [47] reported a significant downregulation of vinculin by real-time reverse transcription PCR (qRT-PCR) and Western Blotting in five invasive human colon cancer cell lines (HCT116, Caco2, HT29, SW620 and SW480) and tissue samples from patients with metastatic colon cancer. An early study also demonstrated a low level of vinculin in invasive and metastatic squamous cell carcinomas tissue samples and their matched metastatic samples [50]. Consistent with this, our results showed that vinculin expression was markedly reduced in 3D NOR-10 and co-cultured lung spheroids compared to 2D cultures. This indicates that culturing cancer cells within a scaffold functionalised with laminin and fibronectin adhesion peptide sequences enhanced cell motility, which is a characteristic of cancer cells in vivo. In sequence, vinculin expression was markedly reduced in cells grown in Fmoc-SAP hydrogels and vinculin expression was higher in cells cultured in 2D and 2.5D as they are less motile and more adherent to artificial tissue culture plastic and to each other, respectively.

The interaction of cells with the surrounding ECM requires cellular transmembrane linkers, such as integrins, to facilitate the adhesion of ECM proteins, including laminin and fibronectin, to the cell’s cytoskeleton (in particular, the F-actin) [51]. Following interaction of integrins with the ECM, focal adhesion complexes are formed comprising of many cytoplasmic proteins including vinculin [51]. When cells interact with the surrounding ECM, they become motile [52]. Therefore, to further investigate cell adhesion and migration, as well as changes in the cytoskeleton organisation in response to the microenvironment, 3D co-culture spheroids were stained for vinculin (one of the focal adhesion markers) and F-actin, and results were compared to 2.5D co-culture spheroids and co-culture cells in 2D cell culture (Figure 3C) over 72 h. Our results showed dramatic changes in filamentous actin distribution in co-culture spheroids when cultured on different cell culture substrates. In 3D cultures, co-cultured spheroids displayed round branched F-actin filaments that varied markedly from the strikingly flat stretched out F-actin filaments of co-cultured cell in 2D cell culture (Figure 3C, arrow) and the round F-actin filaments localizing along the cell periphery in 2.5D spheroids. Similar findings were observed by Zhou et al. [53] where mesenchymal stem cells (MSCs) displayed stretched out F-actin filaments in 2D cell culture and cortical F-actin filaments around cell membranes in spheroids. Therefore, the mechanical properties of cell culture substrates govern the shape and behaviour of cells, which ultimately influence the structure of the cytoskeleton. Distinct focal adhesion points localising outside the plasma membrane was prominently absent in all co-culture conditions. However, cytoplasmic vinculin was clearly observed in each condition [53]. Our results highlight the impact of the ECM on actin cytoskeleton organisation and suggest that functionalised Fmoc-SAP hydrogels enhance the adhesion and migratory phenotype of the co-cultured 3D spheroids.

### 3.4. Functionalised Fmoc-SAP Hydrogel Facilitates Lung Cancer Cell Migration

During cancer progression, some epithelial cancers undergo epithelial-to-mesenchymal transition (EMT), where they switch from a more adherent phenotype to a more migratory metastatic mesenchymal phenotype [54]. During EMT, epithelial markers such as E-cadherin are downregulated, whereas mesenchymal markers such as α-SMA, an actin isoform, are upregulated [54,55]. α-SMA is normally expressed by differentiating fibroblasts to cancer-associated fibroblasts (CAFs) in the TME. In this study, α-SMA expression was determined to assess the differentiation of fibroblasts into CAFs and induction of EMT in SAP hydrogel-encapsulated spheroids by flow cytometry (Figure 3A, Appendix A). LLC, NOR-10 and co-cultured 3D spheroids were encapsulated within the Fmoc-SAP hydrogel and following incubation for 72 h, α-SMA expression was determined by flow cytometry. Non-encapsulated 2.5D spheroids were used as a control and results were also compared to cells cultured in 2D. Results showed a reduction in the expression of α-SMA in NOR-10 cells compared to LLC cells and co-culture in all conditions, suggesting reduced fibroblast activation in the absence of cancer cells (Figure 3A). However, a significantly greater percentage (36.3%) of cells in 3D co-cultured spheroids expressed α-SMA when encapsulated within Fmoc-SAP hydrogel compared to non-encapsulated 2.5D spheroids and 2D co-cultures (*p* < 0.0001 in both conditions). Interestingly, LLC spheroids in 3D and 2.5D also expressed significantly higher percentages (21.1% and 22.9%, respectively) of α-SMA compared to LLC in 2D culture, indicating that the 3D and 2.5D LLC spheroids may have undergone EMT (*p* = 0.0005 and 0.0001, respectively).

In this study, the results showed that embedding cancer cells within 3D functionalised Fmoc-SAP hydrogel not only increased cell proliferation, but also acquired a mesenchymal phenotype by increasing α-SMA expression in LLC and co-cultured spheroids compared to cells cultured in 2.5D and 2D. The 3D functionalised hydrogel provided the cancer cells with mechanical and biochemical signals such as the TME in vivo. A recent study has demonstrated elevated expression of α-SMA along with other mesenchymal markers in liver cancer tissue samples versus normal liver tissues by immunohistochemistry, qRT-PCR and Western blotting [56]. Our results show a general reduction in the expression of α-SMA in NOR-10 fibroblasts compared to LLC and co-culture in all conditions, suggesting reduced fibroblast activation in the absence of cancer cells. Several studies showed that fibroblasts were activated to CAFs following a reciprocal interaction with cancer cells [57,58]. Therefore, our data indicates that cells have undergone EMT after encapsulation within biomimetic Fmoc-SAP hydrogel in vitro.

The aim of study was also to demonstrate that any cellular response within the scaffold is due to the 3D scaffold’s ability to mimic the natural ECM environment via the presence of biological stimuli. To ensure that RGD- and IKVAV-functionalised Fmoc-SAP hydrogels facilitate the migration of cancer cells in co-cultured spheroids by integrin and cancer cell interaction, co-cultured spheroids were tracked. 1% agarose hydrogel provides structure and support to the spheroids; however, it lacks functionalization, and therefore, was again chosen as a control. LLC cells were loaded with the cell CMTPX Dye (Figure 4A, red) and NOR-10 cells were loaded with MitoTracker™ Green FM (Figure 4A, green), and were encapsulated within Fmoc-SAP functionalised hydrogel or 1% agarose hydrogel. After 72 h of incubation, the tracked spheroids were imaged under confocal microscope. Vinculin and α-SMA (Figure 4B) expressions were also measured in co-cultured spheroids encapsulated within both hydrogels by flow cytometry. Our results showed the enhanced motility of CMTPX-labelled LLC and MitoTracker-labelled NOR-10 co-cultured spheroids in Fmoc-SAP hydrogel, whereas spheroids stayed immobile within 1% agarose, lacking the nanofibrous RGD and IKVAV signals. Cancer cells’ motility within Fmoc-SAP hydrogel over non-functionalised 1% agarose was due to the loss of cell adhesion as the vinculin level was significantly downregulated (0%) compared to cells within 1% agarose (23.6%) (Figure 4B). The present findings are in agreement with a previous study that observed the inhibition of U373-MG human glioma spheroid cell migration within 1% agarose compared to collagen and composite collagen-agarose gels [59]. These results highlight the significance of using this model to replicate lung cancer cell adhesion and migration in vitro.

## 4. Conclusions

Growing cancer cell spheroids within a 3D microenvironment provide contextual and chemical cues that a 2D microenvironment lacks—interaction with neighbouring cells, and mechanical and biochemical support from an ECM-like scaffold. These interactions are vital to cellular processes such as cancer cell metabolic activity, differentiation, and morphology. Without these cues (such as in our 2.5D and 2D models), the cumulative effect of the native tumour microenvironment is negated. The encapsulation and growth of a lung tumour spheroid within a programmed RGD/IKVAV peptide-based Fmoc-SAP hydrogel habitat provides a system that furnishes the cells with enough information to mimic the native tumour microenvironment. We have demonstrated that the provision of contextual (spheroid-tumour-form and 3D mechanical support) and chemical (pro-adhesive hydrogel scaffold) signals enables a novel form of biomimetic 3D lung cancer model in vitro. The data revealed that the encapsulation of co-culture spheroids within Fmoc-SAP hydrogels and the interaction of tumour cells and fibroblasts with signalling sequences present on scaffold fibrils enhanced the metabolic activity of tumour spheroids, promoted invasiveness of cancer cells via reductions in vinculin expression, enhanced cellular migration and induced EMT by decreasing alpha-SMA expression. Our results suggest that co-assembled Fmoc-SAP hydrogels functionalised with RGD and IKVAV adhesion sequences are effective 3D cell culture systems in vitro. Furthermore, the encapsulation of co-culture lung tumour spheroids within Fmoc-SAP hydrogels represents an effective lung cancer model in vitro. A key issue with cancer studies is addressing the backlog of candidate drugs without resorting to human or animal trials. This system may be useful as a tool for rapid, in vitro drug screening applications in a low-cost system that recapitulates key features of the natural ECM, particularly for the rapid assessment of novel and repurposed drug candidates. Previously, flow-induced hydrodynamic shear stress has also increased EMT in 2D and 3D lung-cancer models, thus, future investigations may wish to incorporate a more effective model of measuring the 3D migration of the cells, such as a modified transwell system, its development into a bioprinted system [60] or a microfluidic platform to further access cell mobility and drug effectiveness within an artificial TME [61] The precise nature of the migrating cells and the EMT such as features of the system can be further explored via the modular approach we have detailed here. For example, other proteins [62,63] and biopolymers [64] can be coassembled to the system, or a population of healthy cells can be cultured in 3D before the addition of a spheroid. The lung tumour model detailed within this study could be used to further explore effective treatment of lung cancer via more tumour-like conditions, drug delivery and a model of cell mobility.

## Figures and Tables

**Figure 1 gels-08-00332-f001:**
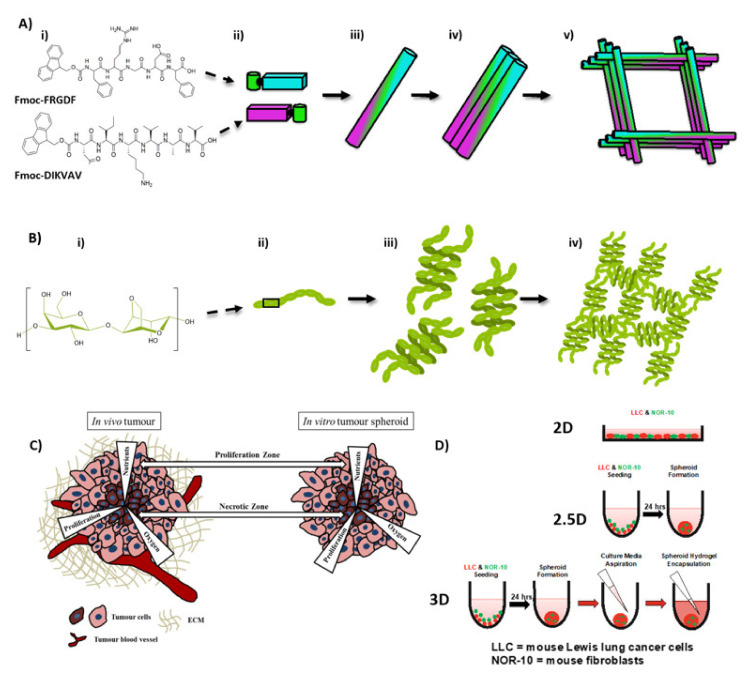
Illustration of Fmoc-SAP and agarose hydrogel self-assembly, and schematic illustration of common features demonstrated by solid tumour in vivo and the in vitro tumour spheroid (**A**) Cartoon depiction of Fmoc-SAP hydrogel formation (i) Each single SAP consists of an Fmoc aromatic group coupled with a short amino acid sequence. (ii) Under physiological conditions, self-assembly is initiated through stacking. (iii) Peptides align to form individual nanofibres. (iv) Nanofibres bundle together. (v) Bundles associate and intertwine to form the hydrogel network. (**B**) Cartoon depiction of agarose hydrogel formation. (i) Chemical structure of agarose. (ii) Artistic depiction of polymer. (iii) Formation of helical structures (iv) Polymer association and formation of network. (**C**) Common features between solid tumours in vivo and tumour spheroids in vitro include a central necrotic zone, due to the decreased availability of oxygen, nutrients and space for proliferation, and an outer proliferation zone. (**D**) Schematic representation of cell growth in the three microenvironments investigated: as a monolayer in 2D tissue culture plates, 2.5D spheroid growth in culture media (i.e., no hydrogel encapsulation), and 3D spheroid growth encapsulated within Fmoc-FRGDF/Fmoc-DIKVAV hydrogels. Common features between solid tumours in vivo and tumour spheroids in vitro include a central necrotic zone, due to the decreased availability of oxygen, nutrients and space for proliferation, and an outer proliferation zone.

**Figure 2 gels-08-00332-f002:**
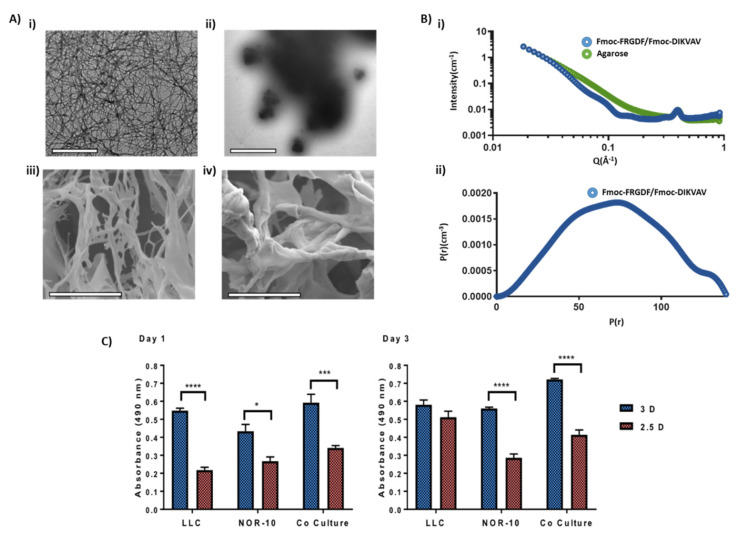
Material characterisation and assessment of cellular metabolic activity. (**A**) TEM and CryoSEM analysis of 1% agarose and Fmoc-SAP co-assembled hydrogels. Top: TEM analysis of (i) Fmoc-FRGDF/Fmoc-DIKVAV co- assembled hydrogel and (ii) Agarose hydrogel (scale = 1 μm). Bottom: CryoSEM analysis of (iii) 1% agarose and (iv) Fmoc-SAP co-assembled hydrogels imaged at 3000× magnification (scale = 20 μm). (**B**) Fitting of scattering curves obtained during SAXS investigation. (i) Fitting for correlation length, and (ii) fitting to two-power model used for the determination of mesh size (arrows indicate the power intercept). (**C**) A comparison of the metabolic activity of LLC, NOR-10 and co-culture 3D and 2.5D spheroids, measured by MTS assay at 490 nm wavelength after an incubation time of 1 and 3 days. Data is presented as mean ± SEM, *n* = 3. Statistics were obtained by two-way ANOVA with the Tukey’s multiple comparisons test. * *p* < 0.05, *** *p* < 0.001, **** *p* < 0.0001. 24 h after seeding, metabolic activity was significantly higher for all cells grown in 3D conditions, and 72 h after seeding, the trend continued for NOR-10 cells and co-cultured cells, while LLC activity was consistent under both conditions.

**Figure 3 gels-08-00332-f003:**
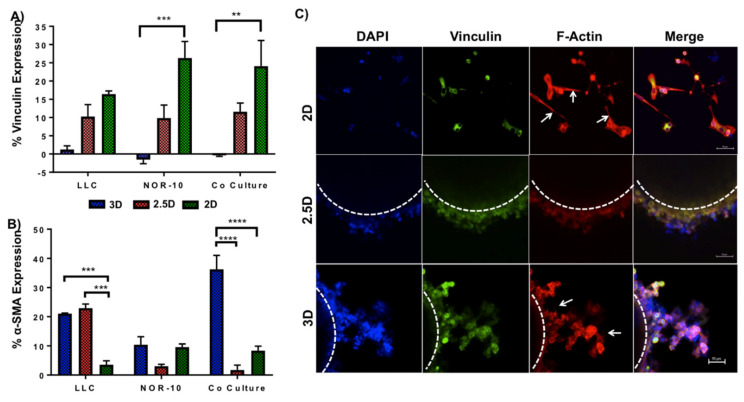
Vinculin expression and co-localisation with F-actin in 3D spheroids encapsulated within Fmoc-SAP hydrogel, 2.5D spheroids and 2D cell culture. (**A**) Flow cytometry analysis of vinculin expression and (**B**) α-SMA expression, both measured in LLC, NOR-10 and LLC+NOR-10 co-culture after 72 h encapsulation within Fmoc-SAP hydrogel (3D) compared to no-encapsulated 2.5D spheroids and 2D cell culture. Data are presented as mean ± SEM, n = 3. Statistics were obtained by two-way ANOVA with the Tukey’s multiple comparisons test. ** *p* < 0.01, *** *p* < 0.001, **** *p* < 0.0001. (**C**) Spheroids were immunofluorescent stained for nucleus (blue), vinculin (green) and F-actin (red) and were analyzed by confocal microscopy following 72 h growth (top to bottom): on 2D tissue culture plates; in media; and in Fmoc-SAP hydrogel. Arrows indicate F-actin filament branching.

**Figure 4 gels-08-00332-f004:**
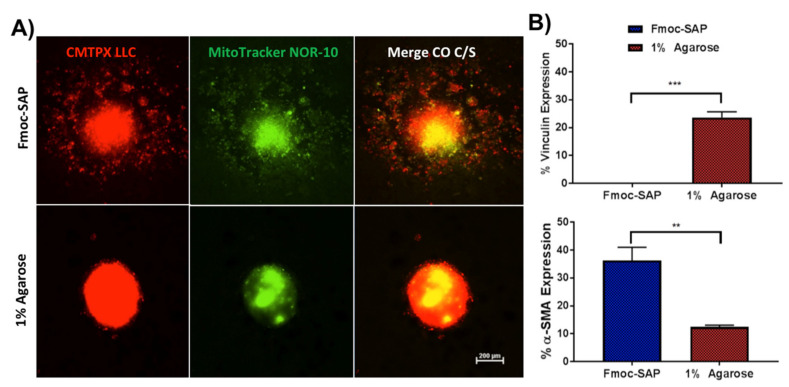
Tracking co-culture lung cancer spheroid migration in functionalised Fmoc-SAP and unfunctionalised 1% agarose hydrogels. (**A**) Migration of co-culture 3D spheroids tracked by labelling LLC cells with CMTPX red dye and NOR-10 with Mitotracker green dye following imaging via confocal microscopy and encapsulated within functionalised Fmoc-SAP hydrogels and unfunctionalised 1% agarose. Merged images are shown in the last column (scale = 200 μm). (**B**) Flow cytometry analysis of vinculin and α-SMA expression in co-culture spheroids encapsulated within Fmoc-SAPs hydrogels and 1% agarose. Data are presented as mean ± SEM, n = 3. Statistics were obtained by two-way ANOVA with the Tukey’s multiple comparisons test. ** *p* < 0.01, *** *p* < 0.001.

## Data Availability

Not applicable.

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
