# Peer review of "Self-Assembled Peptide Habitats to Model Tumor Metastasis"

_gels, 2022, doi:10.3390/gels8060332_

Round 1

Reviewer 1 Report

The aim of this study is to a significantly improved tool, consisting of a novel matrix of functionally programmed peptide sequences, self assembled into a scaffold, is investigated to enable a synthetic model of the growth and migration of multicellular lung tumor spheroids, as proof-of-concept. The study is well organized and the results are clearly presented. The team of authors has worked on a similar issue before. My recommendations are listed below.

  1. Section 2.1: Has the effect of sterilization been investigated?
  2. The aim of the study is very well summarized at the end of the introduction. The tests carried out for this purpose can also be briefly summarized here.
  3. More detailed information is needed regarding vinculin.
  4. The findings obtained as a result of the tests carried out are sufficiently well discussed. However, the contribution and innovative aspect of the results obtained in section 4 (conclusion) can be better emphasized.

Author Response

1: the system is unaffected by UV light as it contains no photocatalytic centers, and the UV process is very mild. 

the text has been modified to read "exposed to UV light in the tissue culture hood.." for clarity

2. A short section has been added at pg 5, ln 12. "The microstructures of the systems were analysed via TEM and Cryo-SEM. Small angle X-ray scattering (SAXS) was used to determine the relative size of the mesh presented by the scaffold features. "

3. the text has been modified to include more detail on Vinculin pg 10, ln 15: " Vinculin is a cytoskeleton protein contributing to cell-cell adhesion through cadherin and cell-ECM connection through integrin receptors.[37] It is closely associated with a role in governing cell-matrix adhesion. It plays a critical role in regulating integrin clustering, and generates the forces required to attach the cells to the ECM. Significant reduction of vinculin levels eventually results in decreased cell adhesion and induction of motility, and can be considered a mechanism for monitoring cancer cell invasion and metastasis.[37]

4: A section has been added to highlight the potential of the system pg 14 ln  "A key issue with cancer studies is addressing the backlog of candidate drugs without resort to human or animal trials. This system may be useful as a tool for rapid, in vitro drug screening applications in a low cost system that recapitulates key features of the natural ECM, particularly for the rapid assessment of novel and repurposed drug candidates:

Reviewer 2 Report

The paper titled "Self-Assembled Peptide Habitats to Model Tumor Spheroid
Models" is well prepared. The topic has enough importance and discussions were carefully presented. All figures are good made to be understanded. Therefore, I suggest this paper can be considered to be accepted. 

Author Response

We thank the reviewer for thier kind comments about our research. 

Reviewer 3 Report

Comments: . The topic is interesting. However there are few  issues should to be addressed 

  • There are few English typos errors should have been revised strongly over all the text.
  • The method by which spheroid 2.5D was encapsulated inside co-assembled Fmoc-SAP hydrogel was not exactly described. Authors should to explain how 2.5D was successfully inserted into hydrogel.
  • The mechanism by which Fmoc-FRGDF and Fmoc-DIKVAV peptides were assembled agarose hydrogel was not clear. It is expected that UV light that was used for sterilization, is initiating crosslinker between agarose and peptide. Authors should to investigate this matter well.
  • The expression of α-SMA was evaluated. However, migrating experiment is recommended either by scratch assay  or by Transwell Cell Migration/Invasion Assay
  • According to Figure 2, co culture exhibited significant proliferation compared to other materials. This indicates that peptide could stimulate cells to  secret   growth factors especially TGF β.  Secreted TGF β should to be investigated in media during cell growth for 27h
  • In figure 4, number of migrating cells was not calculated. however, the fluorescence labelled cells showed many separated cells around spheroid structure. such this condition cannot represent the EMT.  Other experiment should to be used to investigate if these separated cells are related to mechanism of EMT or not.

Author Response

The typographical errors throughout the manuscript have been revised. 

  • The method by which spheroid 2.5D was encapsulated inside co-assembled Fmoc-SAP hydrogel was not exactly described. Authors should to explain how 2.5D was successfully inserted into hydrogel.

2.5D referes to the spheroid cultured in media without encapsulation - i.e. no hydrogel is present. The text has been modifed as follows to clarify this:

figure 1 caption: 2.5D spheroid growth in culture media (i.e. no hydrogel encapsulation),

pg 5, ln 14: "In order to probe the efficacy of the 3D model, we compared outcomes from cells grown within one of three microenvironments, 2D (a monolayer cells grown on flat tissue culture plates), 2.5D (spheroid on the bottom of the well plate, surrounded by media) and 3D (spheroid encapsulated within hydrogel) (Figure 1D). "

  • The mechanism by which Fmoc-FRGDF and Fmoc-DIKVAV peptides were assembled agarose hydrogel was not clear. It is expected that UV light that was used for sterilization, is initiating crosslinker between agarose and peptide. Authors should to investigate this matter well

The peptides were coassembled in a thermodynamic process. we have previously reported on the mechanism that underpins this system. Agarose is used as provided in a thermoset material, used as non-functional control to evaluate the effect of the peptide presentation in the fibrillar structures of the SAP. the peptides were not assembled with the agarose. There is no UV crosslinking involved, as we introduce no photocatalytic centers or reactive groups to the compounds and the UV is under very mild conditions in a culture hood. 

  • The expression of α-SMA was evaluated. However, migrating experiment is recommended either by scratch assay  or by Transwell Cell Migration/Invasion Assay

We thank the reviewer for this insightful comments. However, these assays are suitable for monolayer cultures. We are however, observing the cells migrating out into a 3D system, i.e. in all directions at once. Indeed, the key insight is that the migration occurs once the cells are presented with a functionalised peptide system, and is not just a function of the material encapsulation. We used confocal microscopy to observe this process.

We have modified the text to reflect the need for an effective migration model as follows: pg 14 ln 8 . "Previously, flow-induced hydrodynamic shear stress has also increased EMT in 2D and 3D lung-cancer models, thus, future investigations may wish to incorporate a more effective model of measuring the 3D migration of the cells, such as a modified transwell system or a microfluidic platform to further access cell mobility and drug effectiveness within an artificial TME.[48]"

  • According to Figure 2, co culture exhibited significant proliferation compared to other materials. This indicates that peptide could stimulate cells to  secret   growth factors especially TGF β.  Secreted TGF β should to be investigated in media during cell growth for 27h

the aim of this study was to determine the feasibility of using a novel biomaterial to study the biocompatibility and migratory ability of lung cancer cells. We focused mainly on the mechanical and physical characteristics of the biomatrix.   In future studies, we will study cellular and molecular mechanisms more in depth which will include assays that detect the secretion of growth factors such as TGFb, which will be used as an assay of drug effectiveness (for example)

  • In figure 4, number of migrating cells was not calculated. however, the fluorescence labelled cells showed many separated cells around spheroid structure. such this condition cannot represent the EMT.  Other experiment should to be used to investigate if these separated cells are related to mechanism of EMT or not.

We understand that the reviewer has concerns about the identity of these cells. The purpose of the study is to create a system in 3D that allows cells to migrate within a 3D space in a way that they are unable to in other systems. This gives researchers a hydrogel to characterise the effects of perturbations in this system, and the tools to add further materials, signals and components to the system. We have shown that these gels can be used to culture healthy cells, stem cells and tissue; a spheroid introduced to these cell laden system would allow the researcher to explore the interaction of the 'healthy cells' with the 'tumour cells' from the spheroid. This paper represents a first step toward this.

We have modified the future potential and this concern in the modified text pg 14 ln 12: "The precise nature of the migrating cells and the EMT like features of the system can be further explored via the modular approach we have detailed here. For example, other proteins can be coassembled to the system, or a population of healthy cells can be cultured in 3D before the addition of a spheroid. "

Round 2

Reviewer 3 Report

Authors have revised comments of reviewer point by point and manuscript is acceptable NOW